# Characterization of Commercial Polymer–Carbon Composite Bipolar Plates Used in PEM Fuel Cells

**DOI:** 10.3390/membranes12111050

**Published:** 2022-10-27

**Authors:** Miroslav Hala, Jakub Mališ, Martin Paidar, Karel Bouzek

**Affiliations:** Department of Inorganic Technology, University of Chemistry and Technology, Prague, Technická 5, 16628 Prague, Czech Republic

**Keywords:** PEM, fuel cell, bipolar plate, carbon–polymer composites, material selection, electrical conductivity, mechanical strength

## Abstract

Bipolar plates represent a crucial component of the PEM fuel cell stack. Polymer–carbon composites are recognized as state-of-the-art materials for bipolar plate manufacturing, but their use involves a compromise between electrical and heat conductivity, mechanical strength and costs. Thus, all key parameters must be considered when selecting a suitable plate satisfying the demands of the desired application. However, data relevant to commercial materials for such selection are scarce in the open literature. To address this issue, 13 commercially available polymer–carbon composites are characterised in terms of the following parameters: through-plane conductivity, hydrogen permeability, mechanical strength, water uptake, density, water contact angle and chemical stability. None of the materials tested reached the DOE target for electrical conductivity, while five of the materials met the target for flexural strength. The overall best-performing material showed a conductivity value of 50.4 S·cm^−1^ and flexural strength of 40.1 MPa. The data collected provide important supporting information in selecting the materials most suitable for the desired application. In addition, the key parameters determined for each bipolar plate supply important input parameters for the mathematical modelling of fuel cells.

## 1. Introduction

Proton-exchange membrane (PEM) fuel cells (FCs) play an important role in the transition to a carbon dioxide-neutral economy, including transportation [1]. Combined with electrolysers to generate hydrogen in times of excess renewable energy, fuel cells can be used to bridge temporal and spatial gaps between the availability and consumption of energy [2]. FCs have been extensively investigated over the decades, but the technology has yet to be successfully fully commercialized [3]. This is mainly due to the high cost of FC components, including bipolar plates (BPs). While catalysts and membranes are also costly [4], BPs account for 60–80% of the total weight of an FC stack and 20–30% of its manufacturing cost [5]. Thus, the cost and performance of the FC stack could be considerably improved by selecting the optimal BP material [6,7]. Apart from the cost, the physico-chemical properties of the individual BPs also play an important role in FC stack performance and durability. 

For commercial mass applications, metallic BPs are preferred due to their excellent mechanical strength, as well as high electrical and thermal conductivity [8,9,10]. Moreover, they can be easily formed into the desired shape by hydroforming, hollow embossing or hollow embossing rolling [11,12]. However, their insufficient chemical stability in the FC environment represents a major hindrance. Corrosion products formed on the BP surface, namely metal ions, poison the membrane and the fuel cell anode, thus degrading the stack performance [13,14]. One way to prevent the corrosion of BPs is to safeguard their surface with a protective coating. This, however, entails a drastic increase in price [6,15]. 

Polymer–carbon composites are another group of promising materials for BP manufacturing. These materials are produced by moulding a molten thermoset or thermoplastic polymer binder mixed with one or more carbon-based fillers [16]. Many types of fillers of different shapes and sizes have been tested, including natural and synthetic graphite flakes or lumps, carbon black, graphene, carbon fibres, carbon nanotubes, expandable graphite and expanded graphite [17,18,19,20]. Generally, polymer–carbon composites have a lower density than metallic plates, but they need to be thicker because of their lower mechanical strength, which results in stacks with polymer–carbon BPs having a lower power density. Furthermore, polymer–carbon composites exhibit much lower electrical and heat conductivity due to the content of the non-conductive polymer matrix, the carbon conductivity itself is lower than metal conductivity. Conductive polymers, such as polyaniline, are at present under investigation as potential materials for the coating of metallic BPs [21]. Conversely, polymer–carbon composite-based BPs have superior chemical stability compared to metallic ones. In addition, these types of composite materials are also considered for the manufacture of BPs for vanadium redox flow batteries [22]. 

Various studies have been conducted on the influence of the composition of these materials on their performance. The universally accepted target values of the material parameters set by the US Department of Energy [23,24] are shown in Table 1. The electrical properties of BPs are influenced by the type, orientation, dispersion and content of the filler [16,25]. Mechanical strength is determined by the polymer type, matrix/filler surface interface and filler content [26]. Generally, as the filler content increases, the electrical conductivity of the material increases and its mechanical strength decreases [16,27]. The addition of carbon fibres as a secondary filler can counteract this by enhancing the mechanical strength [28]. Furthermore, the use of different sizes of fillers or of particles with higher aspect ratios reduces the amount of void space between particles, thereby leading to increased electrical conductivity. In particular, nanosized carbon black particles and multiwalled carbon nanotubes have been shown to improve both electrical and mechanical performance. There is a threshold concentration, however, above which the low wettability of the particles leads to poor binding to the polymer matrix [18]. In membranes, similar behaviour can be observed when phase segregation between inorganic and organic phases occurs, which finally leads to the cracking of the membrane [29].

Although there is much data available for experimental, laboratory-prepared materials, the properties of commercial materials are not always available or reliable. For example, De Oliveira et al. [30] collected literature data for 37 different composite materials, only five of which were commercially available. The main source of commercial materials data is technical data sheets. Generally, they only provide a limited set of parameters, without describing the methods used to acquire them. This means that the values reported cannot be verified reproducibly. 

The main aim of the present work is to at least partly bridge this gap and to determine a set of reliable values of important physico-chemical parameters for 13 commercially available polymer–carbon BPs. The focus is not only on electrical conductivity, density and flexural strength but also on permeability to hydrogen, tensile strength, hardness, water contact angle, water uptake and chemical stability (Figure 1). This set of data is unique and is an important aid for selecting the optimal BPs for the desired application and the corresponding stack design, as well as a reliable source of information for groups working on the mathematical modelling of fuel cells and fuel cell stacks. 

## 2. Materials and Methods

### 2.1. Sample Materials

Commercial polymer–carbon composite BPs, made available to the authors for this study, were tested. They are listed in Table 2. The type and content of the binder together with the plate production method and thickness are also provided in the table.

### 2.2. Experimental Methods

#### 2.2.1. Electrical Conductivity

Electrical conductivity was measured using a BP sample in the form of a cylinder of 4.9 mm in diameter. The sample was placed between two copper cylinders 1 cm in diameter and compressed under a defined torque of 140 N·m using a screw mechanism. The copper cylinders were then connected to the MI 3242 microohm meter (METREL, Slovenia) and the ohmic resistance of the samples was measured three times. The thickness of the sample was determined with a micrometer (Mitutoyo, Japan). The conductivity value was then calculated using Equation (1):(1)$$\sigma =\frac{\delta }{R\cdot A}$$
where *σ* indicates the electrical conductivity, *R* stands for ohmic resistance, *δ* is the cylinder height of the sample and *A* represents the cross-section area of the sample. Subsequently, 0.05 mm of material was removed from each side of the sample using 800 sandpaper in order to remove the skin layer differing in composition from the bulk material. Subsequently, the resistance of the ground sample was measured and compared with the samples prior to the removal of the skin layer.

#### 2.2.2. Mechanical Strength

Three parameters were selected to describe the mechanical strength of the sample: flexural strength, tensile strength, and hardness. The measurements were accomplished using a LabTest 5.250SP1-VM (Labortech, Czech Republic) universal testing machine. The dimensions of the samples used for each test are shown in Figure 2.

A three-point bending test was used to determine the flexural strength of the BP. The sample was placed on two supporting pillars, then pressure was applied to the middle of the sample by a universal testing machine. The pressure and distance travelled by the adapter were controlled by a computer. The sample dimensions, collected pressure, travel distance and testing machine configuration were then used to evaluate the flexural strength of the BP.

To measure the tensile strength of the BP, one side of the sample was attached to the fixed arm of the universal testing machine and the other side of the sample was attached to the movable arm. While the arms were moving apart, a camera recorded the elongation of the sample and a computer logged the force needed to tear the sample apart. The cross-section area and the force were used to compute the tensile strength.

The hardness of the samples was measured using the Vickers method. A pyramid-shaped diamond indentor was pressed into the sample applying the force of 0.49 N for 10 s. The indentations were square. The Vickers pyramid number was determined from the diagonal of the indented shape and the force used.

#### 2.2.3. Permeability for Hydrogen

The permeability of BPs for hydrogen was determined using BP samples in the form of disks with a diameter of 25.2 mm. They were prepared from the sample plates by means of a CNC milling machine. The disks were then attached with cyanoacrylate rubber-toughened glue to flat embedded aluminium ring holders as shown in Figure 3.

The scheme of the experimental arrangement used to determine the permeability for hydrogen is shown in Figure 4. The sample was placed in the chamber with one side open to the atmosphere and the second side connected to a reservoir of known volume and with a defined pressure of hydrogen; in our particular case, a reservoir with a volume of 44 cm^3^ and hydrogen pressurised at 500 kPa was used. The pressure difference on the sample was recorded online using Fluke 700G06 manometers (Fluke, Everett, DC, USA) connected to the computer via a NI PCI6704 data acquisition card (National Instruments, Hungary, Austin, TX, USA). Twenty-four-hour data collection was used.

The hydrogen permeability of the sample is then evaluated using Equation (2):(2)$$\ln\left(\frac{p_{2}-p_{1,\;t}}{p_{2}-p_{1,\;0}}\right)=-P\cdot \frac{A}{V\cdot \delta_{M}}\cdot R\cdot T\cdot t,$$
where *p_2_* stands for atmospheric pressure, *p*_1,*t*_ indicates pressure in the reservoir at time *t*, *p*_1,0_ is pressure in the reservoir at time *t* = 0 s, *P* means the permeability here, *A* indicates the area of the sample, *V* stands for the volume of the reservoir, *δ_M_* is the thickness of the sample, *R* is the universal gas constant, and *T* stands for thermodynamic temperature.

#### 2.2.4. Water Uptake

To measure the water uptake, three sets of rectangular samples of 40 mm by 7 mm were cut. The weight of the dry samples was determined using analytical scales (Ohaus, Zurich, Switzerland). In the next step, the samples were left submerged in demineralized water for 168 h at 25 °C, 50 °C, and 80 °C. Finally, the samples were carefully dried using cotton gauze and weighed immediately. The dimensions of the samples were determined using a micrometer (Mitutoyo, Kawasaki City, Japan).

#### 2.2.5. Chemical Stability

Square samples were prepared and weighed. A Fenton solution for an accelerated test simulating the environment of an operating PEM fuel cell [31] containing 2.5 mmol dm^−3^ of Fe^2+^ in the form of FeSO_4_·7H_2_O in 30% H_2_O_2_ was prepared. The samples were submerged in the Fenton solution for 168 h at laboratory temperature. Afterwards, they were washed with demineralized water and left to dry for 168 h at laboratory temperature. Finally, the degraded samples were weighed. An image of the surface and fracture of the samples was captured using a Hitachi S-4700 scanning electron microscope (Hitachi, Hitachi City, Japan).

#### 2.2.6. Contact Angle

A DSA30 drop shape analyser (Krüss, Hamburg, Germany) was used to measure the contact angle of the samples. Three droplets of exact volume were dropped onto the sample surface and their image was taken from the side. Then, the contact angle formed by the droplet and the sample was determined from the images using picture analysis.

## 3. Results and Discussion

The determined values of the parameters studied for the individual bipolar plates under study are summarized in Table 3. In the following text, the individual parameters are described and discussed in more detail.

### 3.1. Electrical Conductivity

Electrical conductivity is one of the critical parameters for BPs. The US Department of Energy has set the target value for electrical conductivity at 100 S·cm^−1^ [23]. In a number of published studies, laboratory-prepared polymer–carbon materials attained the set values [28,32]. However, as shown in Figure 5, none of the commercial materials characterized in our study achieved this target value. Eisenhuth Melange 6 BP with a through-plane conductivity of 40.7 S·cm^−1^ attained the closest value, followed by Eisenhuth PPS and Eisenhuth Melange with 20.4 and 16.0 S·cm^−1^, respectively. Eisenhuth Melange 6 and Eisenhuth Melange contain 15 and 10% of polyethylene binder, respectively, and are produced by extrusion. Eisenhuth PPS contains 13% of polyphenylene sulphide and is produced by compression. Surprisingly, the materials with a higher binder content performed better. Thus, this indicates that the reason for such good performance is the filler and/or optimised filler–binder combination. Unfortunately, no information about the filler type is available from the producers.

As can be seen from the SEM images in Figure 6, the surfaces of the most and least electrically conductive samples are very different. The surface of the Eisenhuth Melange 6 is compact and homogeneous with minor defects visible. On the other hand, the surface of MEGA Extrusion is porous with areas containing different amounts of conductive filler. The reason for the poor performance of MEGA Extrusion BP consists of insufficient carbon filler dispersion and low homogeneity of the resulting material. Moreover, the question of the characteristics of the filler material and its compatibility with the polymer binder remains open.

Another aspect to be considered is the low conductivity skin layer covering the surface of all samples, with the exception of the TF 6 from SGL Carbon manufactured by compressing thin graphite layers. The skin layer appears in the process of extrusion or compression of a homogeneous pre-set polymer–carbon mixture and greatly influences the contact resistance of the materials formed. This effect can best be observed in the Eisenhuth BMA 6 and Eisenhuth Melange samples. For these BPs, the determined bulk conductivity more than doubled from 13.6 to 37.7 S·cm^−1^ and from 16.0 to 37.2 S·cm^−1^, respectively, after the skin layer had been removed. This means that these samples are the second and third most conductive ones behind Eisenhuth Melange 6. With the skin layer removed, the latter reached a conductivity value of 50.4 S·cm^−1^. The skin layer is only a by-product of the manufacturing process. While the effect of the skin layer on the electrical conductivity is great, no effect on the chemical stability of the materials has been observed. It is thus clear that in the cases described, the skin layer needs to be removed prior to the construction of a fuel cell stack.

### 3.2. Mechanical Strength

Flexural strength is another frequently studied parameter and its 2020 and 2025 target values set by US DOE are 25 MPa and 40 MPa, respectively [23,24]. All samples reached the 2020 target value, except for the layered SGL Carbon TF6. On the other hand, only a few samples were able to achieve the 2025 target value. The thickness of the samples generally determines their flexural strength, but other parameters, such as the type of binder or filler, clearly have a greater impact, because there is no correlation in our data between the sample thickness and its mechanical strength. The samples made with polyphenylene sulphide binder, namely Shin Etsu BPB-B350 and Eisenhuth PPS, had the highest flexural strength, 50.2 and 48.3 Mpa, respectively, while having 15% and 13% binder content and were produced by different methods. The thinnest MEGA Extrusion sample, only 1.4 mm thick, had one of the highest flexural strengths, probably due to its high binder content (25%), the highest among the BPs studied. The sample with the highest electrical conductivity, Eisenhuth Melange 6, barely reached the 2025 target with a flexural strength value of 40.1 Mpa.

Fuel cell stacks are not usually under tensile stress. In the case of this particular parameter, some compromise is thus possible, even if other parameters do favour BPs with lower tensile strength for the given application. The tensile strength of the BPs tested correlates well with their flexural strength. The measured flexural and tensile strength values determined are summarised in Figure 7. A similar correlation can be seen between flexural strength and hardness, i.e., the last mechanical parameter tested. Hardness is important mainly during the construction of a stack and during BP milling. Ease of machining can affect the final shape of the BP, which in turn affects the mass transport of the gases from the transport channels into the GDL [33]. The hardness values measured are summarised in Figure 8. SGL Carbon TF6 had the lowest hardness of the materials studied, with Eisenhuth PPS and Eisenhuth Melange 6 being the hardest. The low hardness of the SGL sample is caused by the layered structure of the material and low binder content. Polyethylene sulphide is used as a binder for the hardest material (Eisenhuth PPS) as well as for the material with one of the lowest hardnesses (Shin Etsu BPB 350). The harder material contains 13% of binder and is made by compression, while the softer material contains 15% and is made by extrusion. From the information available, filler properties in combination with the manufacturing process seem to be responsible for the resulting hardness of the material.

As documented by Figure 5 and Figure 9, BP material electrical conductivity and flexural strength are impacted by the binder content in opposite ways. Whereas the flexural strength of BPs increases with increasing binder content, in the case of electrical conductivity the situation is just the contrary. Combining these two aspects, the optimum binder content is localised between 10 and 15 wt.%. If the upper level is exceeded, the conductivity decreases significantly, while the flexural strength no longer improves significantly. Removal of the differently behaving sample values (SGL Carbon TF6 (10)) strengthens the negative correlation between binder content and composite conductivity while diminishing the correlation between binder content and flexural strength. However, this is mainly due to the low number of samples with low binder content. Importantly, other properties, such as homogeneity, binder and filler type, or manufacturing process, can outweigh the influence of binder content to a certain extent. The lack of a specification of the filler used to produce BPs precludes a discussion of the observed dependencies. Nevertheless, other studies published on laboratory-prepared samples conclude that the type of binder has a massive role in the properties of BPs and can outweigh the negative effects of low or high binder content [17,20,25]. The addition of carbon fibres, on the other hand, increases the mechanical strength of the resulting material [28]. The use of carbon particles of different sizes with concentrations under a certain limit can then increase the electrical conductivity of the material [25].

### 3.3. Permeability of Bipolar Plates for Hydrogen

In the case of the permeability of BPs for hydrogen, the primary concerns are safety issues. Hydrogen permeating through a BP is mixed with air/oxygen in the cathode compartment. Under standard conditions, this just results in a fuel loss, as for exceeding the flammability limit of the mixture, the permeation rate would need to be very high. On the other hand, hydrogen may accumulate in the cooling liquid loop and thus form explosive conditions. Despite the potential for mixing with air/oxygen and forming a dangerous mixture composition, that is more probable during maintenance of the cooling circuit. To mitigate these negative impacts, BPs need to have low hydrogen permeability. The DOE has set the target value of the H_2_ permeability coefficient for 2020 and 2025 at 1.3 × 10^−14^ and 2 × 10^−16^ Std cm^3^·s^−1^·cm^−2^·Pa^−1^, respectively [23,24]. This can be converted to moles using the ideal gas law, resulting in 5.3 × 10^−15^ mol·s^−1^·cm^−2^·Pa^−1^ and 8.2 × 10^−17^ mol·s^−1^·cm^−2^·Pa^−1^. Two of the BPs tested satisfied the 2025 target value, and five samples did not reach the 2020 value (see Figure 10). The sample with the lowest hydrogen permeability, MEGA Extrusion, also has the highest binder content. The next two samples with the lowest permeability, by Shin Etsu, are also produced by extrusion and contain 15% of binder. This, therefore, shows that extrusion leads to BPs with lower permeability for hydrogen. On the other hand, samples produced by compression are characterised by higher permeability. That is because the production method influences the internal structure of the BP leading to different permeabilities to gases. Furthermore, it should be borne in mind that our experiments were conducted at 25 °C and 0% RH as opposed to 80 °C and 100% RH prescribed by the DOE. Usually, the permeability of polymer materials increases with increasing temperature and humidity due to the material’s open porosity caused by the water molecules penetrating into it. It may thus be assumed that the permeability values will be even higher under these conditions.

### 3.4. Water Uptake

Significant water absorption would result in swelling of the BPs. Related dimensional changes can mechanically damage the fuel cell stack. Moreover, as discussed above, it may result in increased BP permeability for hydrogen. Figure 11 summarises the water content in the BP samples after 168 h of swelling. For all samples, with only one exception, water content below 1 wt.% was found. In most cases, it is below 0.3 wt.%. The exception is the SGL Carbon TF 6 which absorbed an amount of water close to a third of its original weight. This is due to the layered structure of this sample, which can accommodate water easily (see Figure 12). In the case of Eisenhuth Melange 6, Eisenhuth PPG 86, Eisenhuth PPS and SGL FR 10 we observed an increase in water uptake at higher temperatures. For the other samples, the increase in temperature led to lower water uptake. Since these observations are not related to the type of polymer binder and information on the filler is not available, it is not possible to explain this observation more closely. In general, at higher temperatures the extruded samples performed better, which could be explained by their less porous structure. The effect of the different production methods on the porosity of the samples is also apparent in the result of the hydrogen permeability characterization. For no material under study were sizeable dimensional changes observed, including the SGL Carbon TF 6 sample. In the very last case, it may be concluded that only the empty spaces or pores in the samples were filled with water. This could lead to the damage of a stack, if the stack temperature changes suddenly, e.g., stack freezing. Slightly lower water swelling at 80 °C, in comparison to 50 °C, is due to the evaporation of water from the surface layers in the case of the higher temperature. The differences thus indicate experimental error which was <5 mg weight difference between measurements at 80 °C and 50 °C.

### 3.5. Chemical Stability

The results observed during exposure of the BP samples to the Fenton solution are summarised in Figure 13. The highest susceptibility to degradation was shown by Eisenhuth PPS BP, which lost 2.52 wt.% of its weight, followed by Shin Etsu BPB-B350 with a loss of 0.45 wt.%, and Eisenhuth PP with a loss of 0.28 wt.%. Two materials with the most significant weight loss use polyphenylene sulphide as the binder. Although this binder improves the mechanical strength of the BPs, low stability in an environment with hydroxyl radicals needs to be considered. The other samples did not lose more than 0.13% of their weight, and in some cases no weight loss was observable. Thermoset and fluoropolymer binders had, in general, the lowest weight loss.

SEM images of the surface of the Eisenhuth PPS (the material that degraded the most) samples before and after the degradation process are shown in Figure 14. Due to the polymer binder degradation, the sample surface is primarily covered by the remaining filler material in this case. Shin Etsu BPB-PP had a relatively small weight loss. No surface damage is observable in the SEM image (see Figure 14).

### 3.6. Contact Angle

Hydrophobicity affects the ability of BPs to facilitate the transport of the produced water out of the fuel cell stack. Water droplets will not adhere to a more hydrophobic surface of BP channels and will thus block the flow field channel. Additionally, a more hydrophobic surface is beneficial as it lowers the water uptake by the BP. The summary of the contact angles at the individual BP surfaces is summarised in Figure 15. In general, extruded samples have shown a higher contact angle, i.e., more hydrophobic properties. The two samples with the highest contact angles are made using a fluoropolymer binder, i.e., a highly hydrophobic polymer. Interestingly, SGL Carbon TF6 has the second lowest contact angle, although using a fluoropolymer binder. However, this sample has the lowest binder content that is only distributed between its layers, leaving bare graphite exposed on its surface. The low contact angle of the MEGA compression sample is caused by the inhomogeneous distribution of the binder in the materials and inhomogeneous surface, resulting in its low hydrophobicity.

## 4. Conclusions

Within the framework of this study, the main characteristics of the commercially available BPs composite materials were determined experimentally. Due to the absence of fundamental information on the materials and conditions used during their production, a detailed discussion of the results obtained is not possible. Thus, this work focuses, to a significant degree, on reporting the experimental information obtained. It is possible to conclude that, whereas most materials satisfy DOE targets for the year 2020, there is plenty of scope left to satisfy the parameters set for the year 2025 (see Table 4). Whereas materials well comply with the targets of flexural strength and hydrogen permeability, the critical aspect seems to be electrical conductivity. Currently, of the commercial polymer–carbon composites studied, Eisenhuth Melange 6 outperformed the other samples in terms of electrical conductivity and had satisfactory results in all other categories. Nevertheless, it is difficult to generalise the outcomes of this research. Specific applications may define particular requirements and different BP materials may prove to be the optimal ones. However, BPs are not used only in fuel cells, but also in redox flow batteries where the importance of various technical parameters is different. The data summarised here aim to provide good guidelines for making a selection, as well as reliable input data for mathematical modelling which constitutes an unavoidable part of industrial cells and cell stack design.

## Figures and Tables

**Figure 1 membranes-12-01050-f001:**
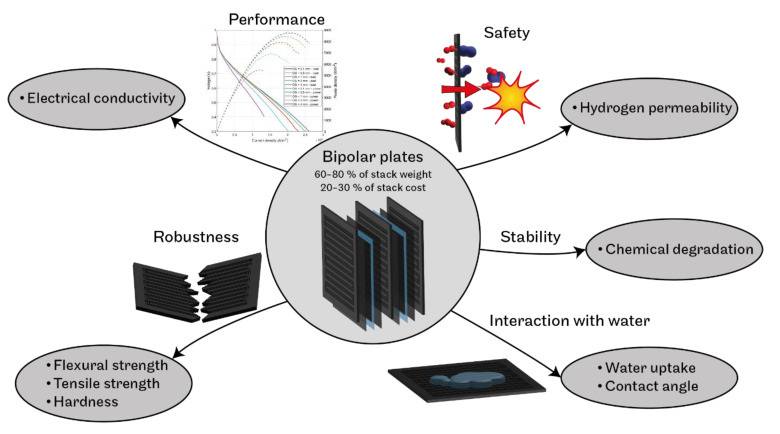
Studied BP parameters and the related aspects.

**Figure 2 membranes-12-01050-f002:**
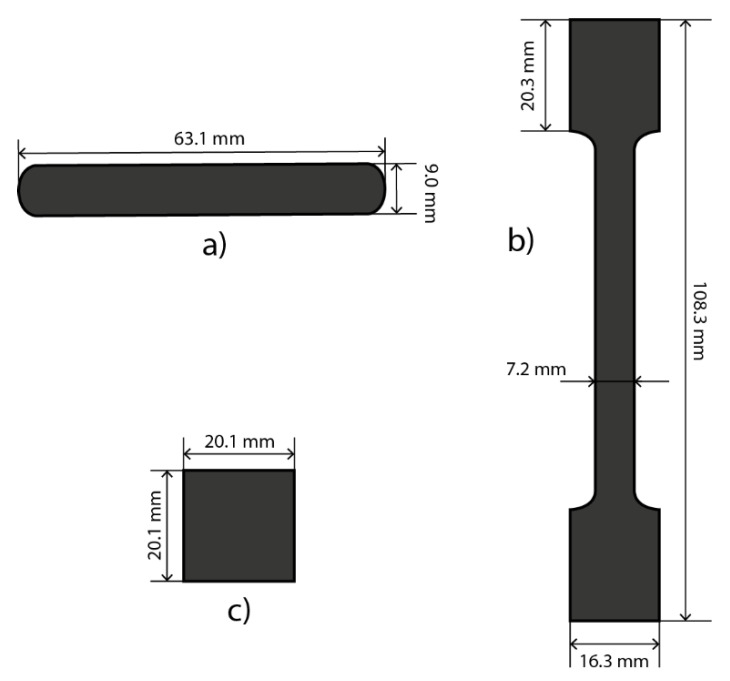
Dimensions of the samples for the measurement of (**a**) flexural strength, (**b**) tensile strength, and (**c**) hardness.

**Figure 3 membranes-12-01050-f003:**
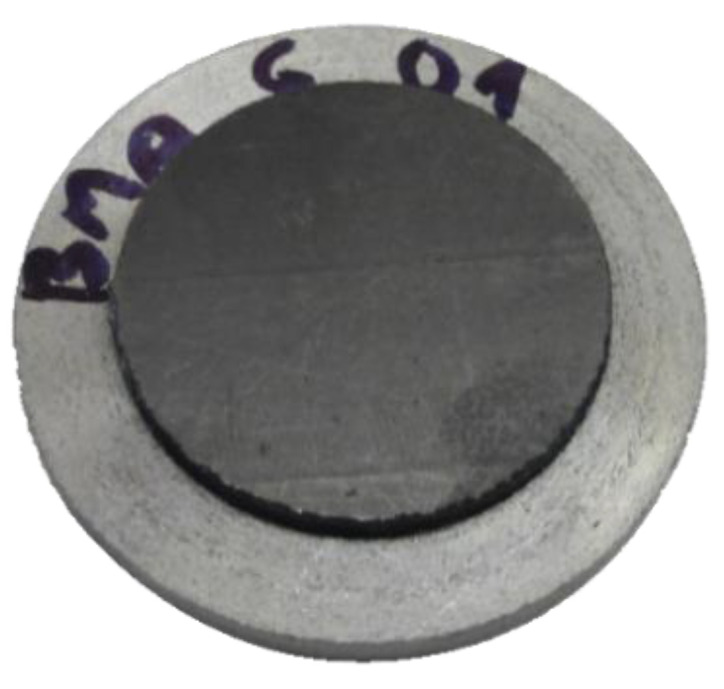
Image of a sample used for evaluation of the BP material’s permeability for hydrogen.

**Figure 4 membranes-12-01050-f004:**
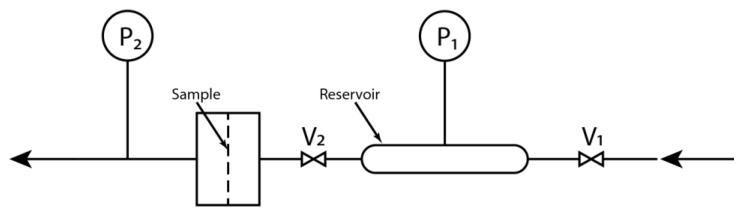
Scheme of the apparatus used to determine BP material’s permeability for hydrogen. Gas is filled into the reservoir by valve V_1_. Then, the reservoir is connected by opening valve V_2_ to the chamber with the sample. The pressure sensor P_1_ records the pressure in the reservoir. The pressure sensor P_2_ measures the atmospheric pressure.

**Figure 5 membranes-12-01050-f005:**
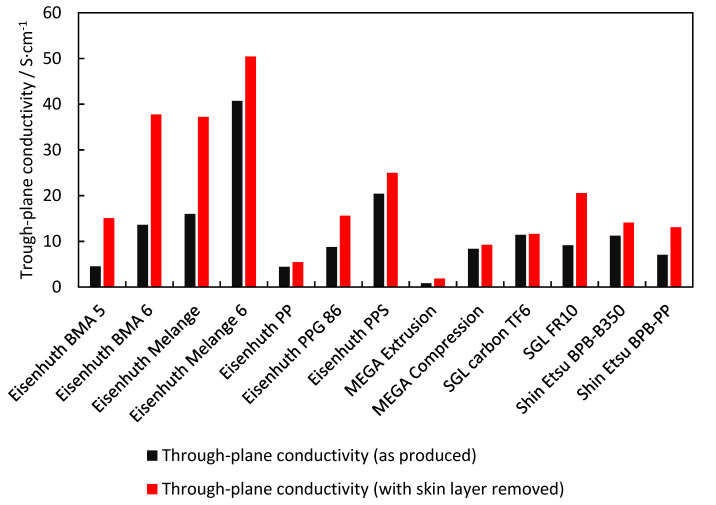
Through-plane electrical conductivity of the BPs tested.

**Figure 6 membranes-12-01050-f006:**
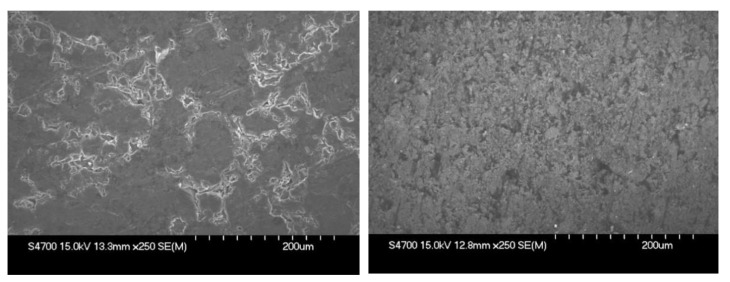
SEM images of the surface of Eisenhuth Melange 6 (**left**) and MEGA Extrusion (**right**).

**Figure 7 membranes-12-01050-f007:**
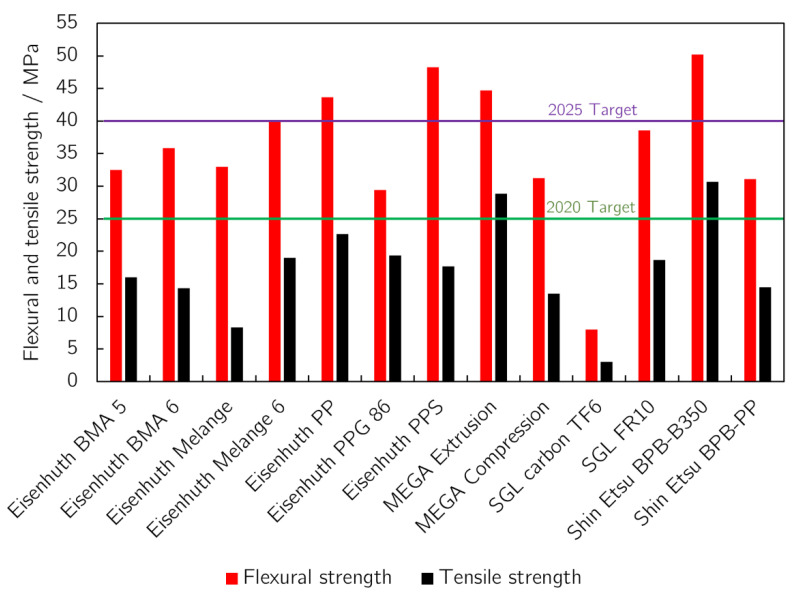
Flexural and tensile strength of the BPs tested. DOE 2020 and 2025 target values [23,24] for flexural strength are shown as green and purple lines, respectively.

**Figure 8 membranes-12-01050-f008:**
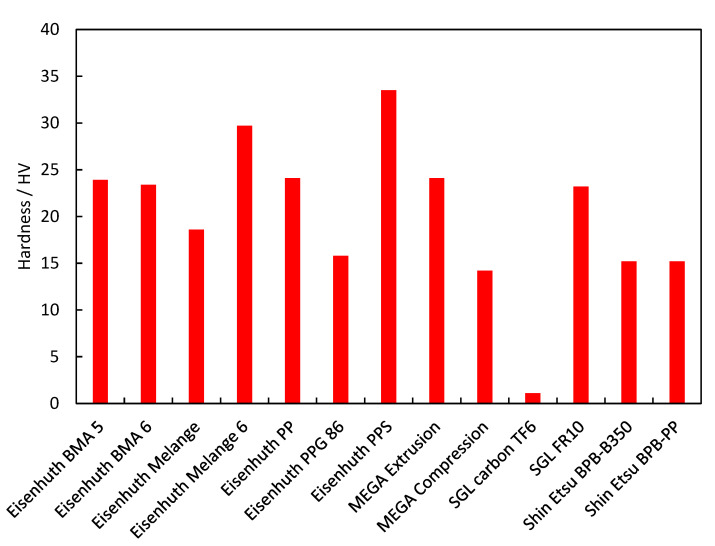
Hardness of the BP materials tested.

**Figure 9 membranes-12-01050-f009:**
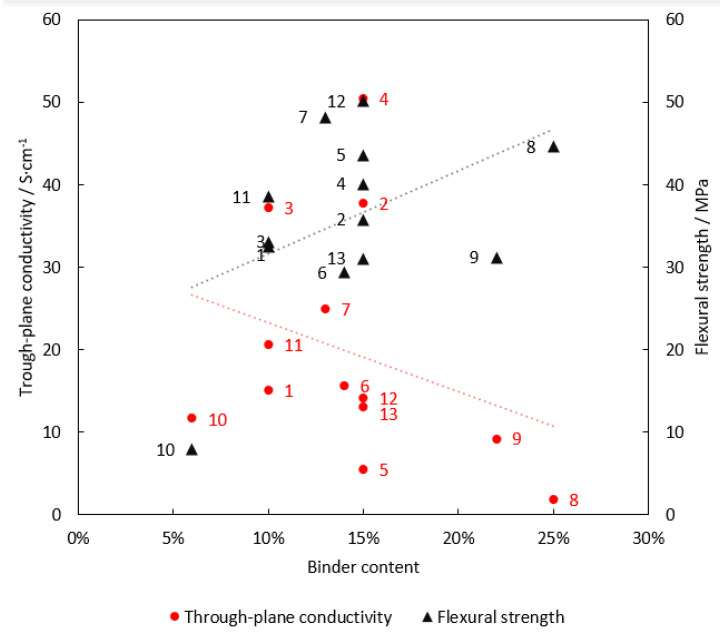
Through-plane conductivity and flexural strength vs. binder content. (1) Eisenhuth BMA 5, (2) Eisenhuth BMA 6, (3) Eisenhuth Melange, (4) Eisenhuth Melange 6, (5) Eisenhuth PP, (6) Eisenhuth PPG 86, (7) Eisenhuth PPS, (8) MEGA Extrusion, (9) MEGA Compression, (10) SGL Carbon TF6, (11) SGL FR10, (12) Shin Etsu BPB-B350, (13) Shin Etsu BPB-PP.

**Figure 10 membranes-12-01050-f010:**
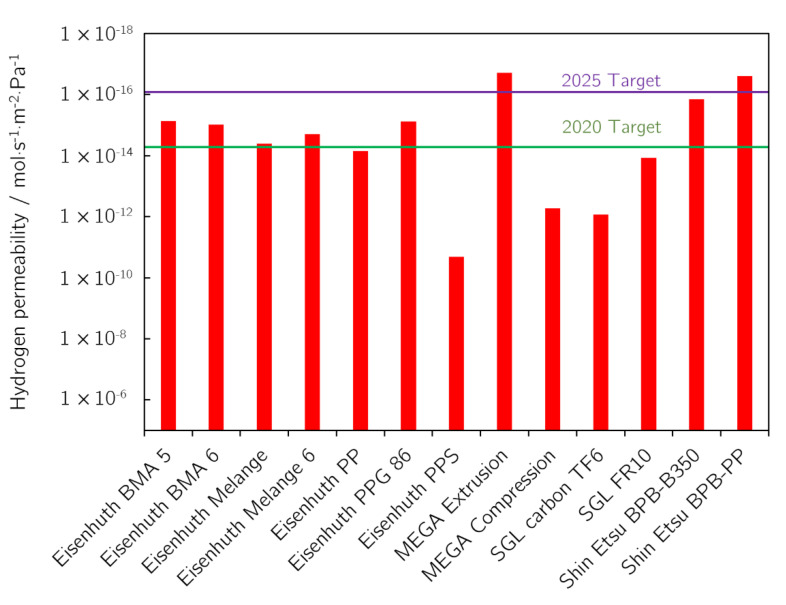
Hydrogen permeability values for BPs tested. DOE 2020 and 2025 target values [23,24] for hydrogen permeability are shown as green and purple lines, respectively.

**Figure 11 membranes-12-01050-f011:**
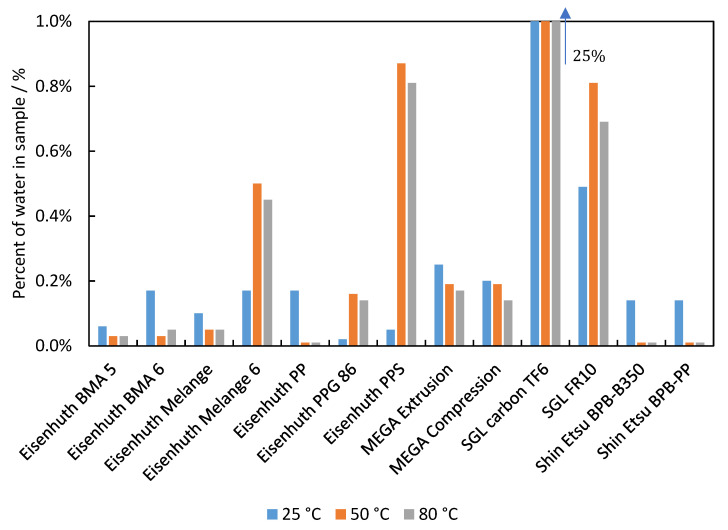
Weight percent of water in the samples after seven days’ swelling at 25, 50, and 80 °C.

**Figure 12 membranes-12-01050-f012:**
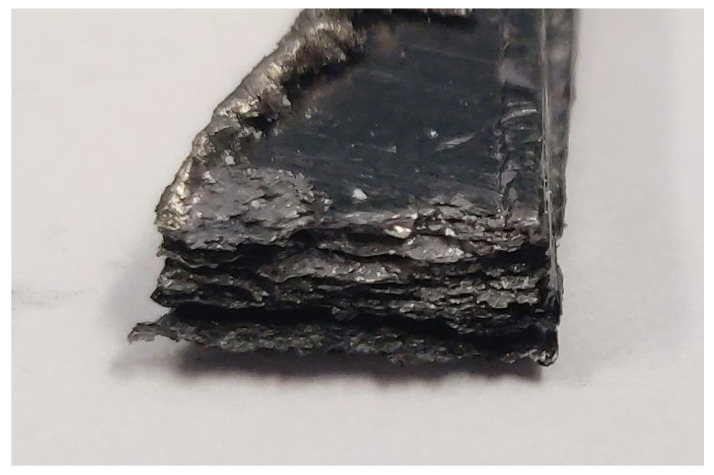
Image of the SGL Carbon TF 6 sample with the layered structure clearly visible.

**Figure 13 membranes-12-01050-f013:**
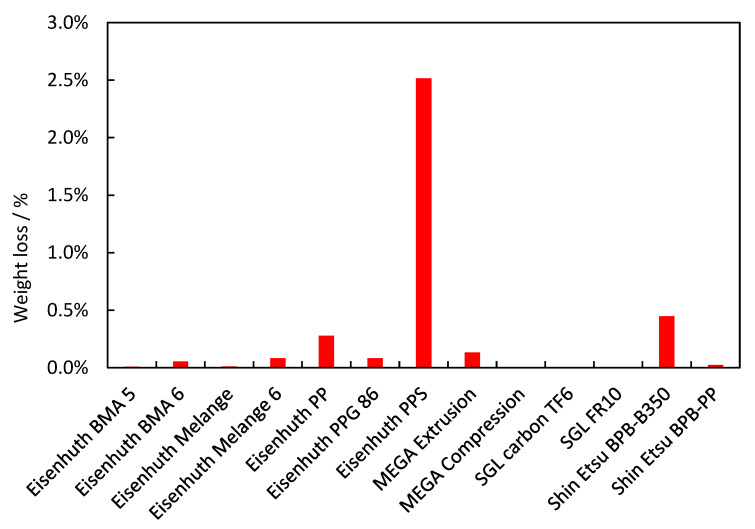
Chemical degradation extent of the BPs tested.

**Figure 14 membranes-12-01050-f014:**
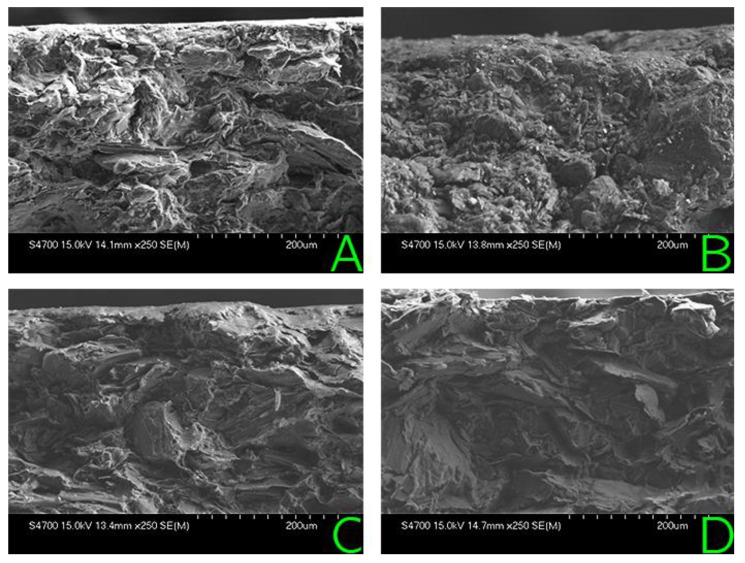
SEM images of the fracture plane of Eisenhuth PPS before chemical degradation (**A**), after chemical degradation (**B**), Shin Etsu BPB-PP before chemical degradation (**C**) and after chemical degradation (**D**).

**Figure 15 membranes-12-01050-f015:**
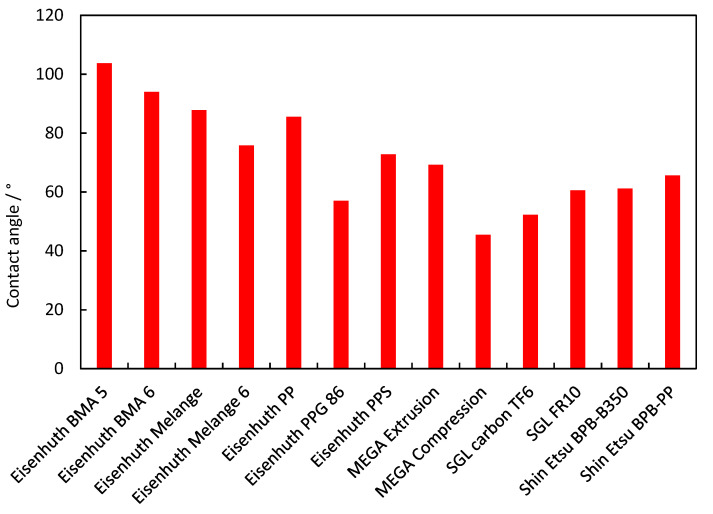
Contact angle of the individual BP materials under study.

**Table 1 membranes-12-01050-t001:** U.S. Department of Energy technical targets for BPs for transportation applications [23,24].

Characteristics	Units	2015 Status	2020 Targets	2025 Targets
Cost	USD·kW_net_^−1^	7	3	2
Plate weight	kg·kW_net_^−1^	<0.4	0.4	0.18
Plate H_2_ permeation coefficient	Std cm^3^·s^−1^·cm^−2^·Pa @ 80 °C, 3 atm, 100% RH	0	<1.3 × 10^−14^	2 × 10^−16^
Corrosion anode	µA·cm^−2^	no active peak	1 and no active peak	<1 and no active peak
Corrosion cathode	µA·cm^−2^	<0,1	<1	<1
Electrical conductivity	S·cm^−1^	>100	100	>100
Area-specific resistance	Ω·cm^2^	0.006	0.01	<0.01
Flexural strength	MPa	>34 (carbon plate)	>25	>40

**Table 2 membranes-12-01050-t002:** List of carbon–polymer composite bipolar plates used in the study, together with their producers, commercial name and main characteristics.

Manufacturer	Name	Binder Proportion	Manufacturing Process	Thickness/mm
Eisenhuth	BMA 5	10% fluoropolymer	extrusion	4.34
Eisenhuth	BMA 6	15% fluoropolymer	extrusion	3.87
Eisenhuth	Melange	10% polyethylene	extrusion	3.80
Eisenhuth	Melange 6	15% polyethylene	extrusion	3.74
Eisenhuth	PP	15% polypropylene	extrusion	5.73
Eisenhuth	PPG 86	14% polypropylene	extrusion	2.04
Eisenhuth	PPS	13% polyphenylene sulfide	compression	3.91
MEGA	Extrusion	25% polypropylene derivative	extrusion	1.44
MEGA	Compression	22% polypropylene derivative	compression	5.04
SGL	carbon TF6	6% fluoropolymer	compression	3.10
SGL	FR10	10% thermoset	compression	3.57
Shin Etsu	BPB-B350	15% polyphenylene sulfide	extrusion	3.04
Shin Etsu	BPB-PP	15% polypropylene	extrusion	3.03

**Table 3 membranes-12-01050-t003:** General overview of the individual bipolar plates’ parameters determined.

Material	Thickness/mm	Conductivity (as produced)/S·cm^−1^	Conductivity (skin layer removed)/S·cm^−1^	Hydrogen permeability/mol·s^−1^·m^−2^·Pa^−1^	Flexural strength/MPa	Tensile strength/MPa	Hardness/HV	Water uptake 25 °C/wt.% H_2_O	Water uptake 50 °C/wt.% H_2_O	Water uptake 80 °C/wt.% H_2_O	Density/kg·m^−3^	Degradation (weight loss)/%	Contact angle /°
2025 DOE Targets [23]	-	100	100	<2 × 10^−16^	40	-	-	-	-	-	-		-
Eisenhuth BMA 5	4.338	4.6	15.1	7.4 × 10^−16^	32.5	16	23.9	0.06	0.03	0.03	1822	0.01	103.7
Eisenhuth BMA 6	3.874	13.6	37.7	9.8 × 10^−16^	35.8	14.3	23.4	0.17	0.03	0.05	2046	0.05	94
Eisenhuth Melange	3.802	16	37.2	4.0 × 10^−15^	33	8.3	18.6	0.10	0.05	0.05	2069	0.01	87.8
Eisenhuth Melange 6	3.741	40.7	50.4	2.0 × 10^−15^	40.1	19	29.7	0.17	0.50	0.45	2020	0.08	75.8
Eisenhuth PP	5.728	4.4	5.5	6.9 × 10^−15^	43.7	22.7	24.1	0.17	0.01	0.01	1985	0.28	85.5
Eisenhuth PPG 86	2.044	8.7	15.6	7.7 × 10^−16^	29.4	19.4	15.8	0.02	0.16	0.14	1800	0.08	57
Eisenhuth PPS	3.911	20.4	25	2.1 × 10^−11^	48.3	17.7	33.5	0.05	0.87	0.81	1946	2.52	72.8
MEGA Extrusion	1.439	0.8	1.8	1.9 × 10^−17^	44.7	28.9	24.1	0.25	0.19	0.17	1487	0.13	69.2
MEGA Compression	5.035	8.4	9.2	5.4 × 10^−13^	31.2	13.5	14.2	0.20	0.19	0.14	1565	0.00	45.5
SGL Carbon TF6	3.086	11.5	11.6	8.5 × 10^−13^	8	3	1.1	24.01	24.39	23.74	957	0.00	52.3
SGL FR10	3.569	9.1	20.6	1.2 × 10^−14^	38.6	18.7	23.2	0.49	0.81	0.69	1976	0.00	60.6/ 21.9
Shin Etsu BPB-B350	3.036	11.2	14.1	1.4 × 10^−16^	50.2	30.7	15.2	0.14	0.01	0.01	1900	0.45	61.2
Shin Etsu BPB-PP	3.034	7.1	13.1	2.5 × 10^−17^	31.1	14.5	15.2	0.14	0.01	0.01	1740	0.02	65.6

**Table 4 membranes-12-01050-t004:** Percent of satisfaction with the 2020 and 2025 DOE Targets.

	2020 electrical conductivity satisfaction /%	2025 electrical conductivity satisfaction /%	2020 hydrogen permeability satisfaction /%	2025 hydrogen permeability satisfaction /%	2020 flexural strength satisfaction /%	2025 flexural strength satisfaction /%
2025 DOE Targets	100		<5.3 × 10^−15^		>25	
2025 DOE Targets		100		<8.2 × 10^−17^		>40
Eisenhuth BMA 5	15	15	106	94	130	81
Eisenhuth BMA 6	38	38	105	93	143	90
Eisenhuth Melange	37	37	101	89	132	83
Eisenhuth Melange 6	50	50	103	91	160	100
Eisenhuth PP	6	6	99	88	175	109
Eisenhuth PPG 86	16	16	106	94	118	74
Eisenhuth PPS	25	25	75	66	193	121
MEGA Extrusion	2	2	86	76	179	112
MEGA Compression	9	9	117	104	125	78
SGL Carbon TF6	12	12	85	75	32	20
SGL FR10	21	21	98	87	154	97
Shin Etsu BPB-B350	14	14	111	99	201	126
Shin Etsu BPB-PP	13	13	116	103	124	78

## Data Availability

Data is contained within the article.

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
