# Peer review of "Characterization of Commercial Polymer–Carbon Composite Bipolar Plates Used in PEM Fuel Cells"

_membranes, 2022, doi:10.3390/membranes12111050_

Round 1
Reviewer 1 Report
Comments on Manuscript.
Manuscript number: membranes-1966373
The article entitled "Characterization of Commercial Polymer-carbon Composite Bi- 2polar Plates Used in PEM Fuel Cells " has been reviewed. My comments and recommendations about the recent article can be found in below. The paper can be accepted.
In General;
* The subject of current investigation and method carried out are within the scope of membranes.
* The topic of Bipolar plates represent a crucial component of the PEM fuel cell stack is quite interesting and the obtained results would be very useful.
* The quality of English grammar is quite satisfactory.
* The obtained results can be very useful for scientific community.
As per my opinion I have drawn the following views given as below:-
1. Originality:- The running text in present article has shown sufficient novelty and excellent interest to warrant for final publication.
2. Results and discussion:- The way of explanation in this part is found very much strong, and I am fully agreed and results presented in this article are in enough satisfaction.
3. Conclusion:- Excellent
Overall, the presentation of work is quite well and satisfactory. The outcomes of the study will add new information to the scientific literature and will be very helpful for the forthcoming workers of the material science.
Author Response
Above all, the authors would like to express thanks and to appreciate reviewer for carefully reading this paper and for pointing out changes, which resulted in an overall improvement of this manuscript.
We thank to the referee for his positive evaluation of our article.
Reviewer 2 Report
This paper characterizes the commercial polymer carbon composite bipolar plate for PEM fuel cell. The research content is very interesting, which can get the attention of researchers and is worth publishing.
However, in order to meet the high quality publishing requirements of journals, please consider the following suggestions.
1) The abstract needs quantitative data and core conclusions.
2) In the first section, it is better to add a figure to explain the research background and motivation.
3) Please pay attention to the typesetting of the first paragraph of Section III.
4) The depth of analysis in the result section is not enough, such as whether the experimental data needs to be combined with the paper of the simulation model?
Relevant references are as follows:
Review: Modeling and Simulation of Membrane Electrode Material Structure for Proton Exchange Membrane Fuel Cells August 2022Coatings 12(8):1145 Follow journal DOI: 10.3390/coatings12081145
5) The discussion needs to be a separate section.
6) The conclusion needs to be rewritten, with tabular views and important data.
Author Response
Above all, the authors would like to express thanks and to appreciate reviewer for carefully reading this paper and for pointing out changes, which resulted in an overall improvement of this manuscript.
With respect to the reviewer comments:
- The abstract needs quantitative data and core conclusions.
Ad1. the abstract was modified and quantitative data were added
- In the first section, it is better to add a figure to explain the research background and motivation.
Ad 2. We thank to reviewer for this point Figure 1. was added for clarification
- Please pay attention to the typesetting of the first paragraph of Section III.
Ad 3. The whole text was again checked by native speaker and found misteakes were corrected
- The depth of analysis in the result section is not enough, such as whether the experimental data needs to be combined with the paper of the simulation model?
Relevant references are as follows:
Review: Modeling and Simulation of Membrane Electrode Material Structure for Proton Exchange Membrane Fuel Cells August 2022Coatings 12(8):1145 Follow journal DOI: 10.3390/coatings12081145
Ad. 4. The missing reference was incorporated to the text with comments.
- The discussion needs to be a separate section.
Ad 5. We think about reviewer recommendation but to prevent repeating of several parts of text we decided to keep results and discussion as one chapter for easier understanding.
- The conclusion needs to be rewritten, with tabular views and important data.
Ad. 6. The conclusion was extended by tabular summary of results obtained.
Reviewer 3 Report
The manuscript reported the comprehensive characterizations of polymer-carbon composite bipolar plates for PEMFC applications. In detail, 13 commercially available polymer-carbon composite bipolar plates are characterized in terms of through-plane conductivity, hydrogen permeability, mechanical strength, chemical stability, etc. This manuscript provides critical references for selecting materials and the key parameters determined may supply important input parameters for mathematical modelling.
The content of this manuscript meets the reading interests of the readers of the journal. However, there are certain English spelling and grammar issues, and also the discussion and explanation should be further improved. I suggest giving a minor revision and the authors need to clarify some issues or supply some more experimental data to enrich the content.
1. For grammar issues, it is suggested that the author double-check the small grammar errors in the full text, especially the lack of and redundant use of definite articles.
2. For the Keywords, ‘materials selection’, ‘electrical and heat conductivity’, and ‘mechanical strength’ should be added in order to attract a broader readership.
3. Page 1, Abbreviations of PEM shall be interpreted;
‘PEM fuel cells (FCs) play an important role in the transition to a carbon dioxide neutral economy’, but where do the hydrogen sources come from? It may combine with the electrolyzers to generate hydrogen by water splitting, and the energy for water splitting can be provided by solar and wind energies, since most renewable energy sources are intermittent, opening spatial and temporal gaps between the availability of the energy and its consumption by the end-users (10.1016/j.electacta.2019.03.056). Excess renewable energy can be stored in the form of hydrogen energy and converted into electric energy through FC when needed.
‘This is mainly due to the high cost of FC components, including bipolar plates (BPs).’ It is also well-known that the cost of catalysts and membranes is also very high (10.1016/j.rser.2015.07.157). I suggest adding some comparisons between them to highlight the significance of this work.
4. Page 2, ‘Furthermore, polymer-carbon composites exhibit much lower electrical and heat conductivity due to the content of the non-conductive polymer matrix...’ Why do not adopt conducting polymers, such as polyanilines? I consider it may make the polymer-carbon composites have similar, or at least not much lower conductivity compared to metallic ones.
5. Page 2, ‘There is a threshold concentration, however, above which the low wettability of the particles leads to poor binding with the polymer matrix [16].’ Why does wettability have a relation with polymer-filler binding? For membranes, there are also inorganic filler-organic polymer hybrid ones, and extremely high loading of fillers also leads to the cracking of the composites (10.1016/j.electacta.2021.138133). This is mainly due to the phase segregation between the organic and filler phases that finally leads to cracking, but not related to wettability since both hydrophobic and hydrophilic fillers demonstrated the same trend.
6. Page 10, ‘It is thus clear that in the cases described, the skin layer needs to be removed prior to the construction of a fuel cell stack.’ In terms of conductivity, it is true that it is better to remove the skin layer before use in a fuel cell stack. But there should be some reasons for using that skin layer, for example, to improve the chemical stability of the BPs by sacrificing a bit the conductivity. I consider there should be some comprehensive discussions about whether it is suitable to really remove the skin layer by considering multiple influencing factors.
7. Page 10, ‘Polyethylene sulphide is used as a binder for the hardest material (Eisen-huth PPS) as well as for the material with one of low hardness (Shin Etsu BPB 350). The harder material contains 13 % of binder and is made by compression, while the softer material contains 15 % and is made by extrusion. From the information available, filler properties seem to be responsible for the resulting hardness of the material.’ I cannot fully agree with these descriptions. I consider that it should be the manufacturing process and binder composition together to determine the hardness. It is too absolute to say that only filler properties is responsible for the hardness, just because one binder leads to both the highest hardness and lowest hardness composites. It must be noted that the treatment/preparation process is also different.
8. Page 13, ‘Figure 9: Hydrogen permeability values for BPs tested. DOE 2020 and 2025 target values [20,21] for flexural strength are shown as a green and a purple line, respectively’. Why does flexural strength appear in a figure related to hydrogen permeability?
Author Response
Above all, the authors would like to express thanks and to appreciate reviewer for carefully reading this paper and for pointing out changes, which resulted in an overall improvement of this manuscript.
With respect to the reviewer comments:
need to clarify some issues or supply some more experimental data to enrich the content.
- For grammar issues, it is suggested that the author double-check the small grammar errors in the full text, especially the lack of and redundant use of definite articles.
Ad. 1. We thank to reviewer for this comment. The text was checked again by native speaker and identified errors were corrected.
- For the Keywords, ‘materials selection’, ‘electrical and heat conductivity’, and ‘mechanical strength’ should be added in order to attract a broader readership.
Ad. 2. Keywords were extended by recommend words. We agree it reflect the topic of article.
- Page 1, Abbreviations of PEM shall be interpreted;
‘PEM fuel cells (FCs) play an important role in the transition to a carbon dioxide neutral economy’, but where do the hydrogen sources come from? It may combine with the electrolyzers to generate hydrogen by water splitting, and the energy for water splitting can be provided by solar and wind energies, since most renewable energy sources are intermittent, opening spatial and temporal gaps between the availability of the energy and its consumption by the end-users (10.1016/j.electacta.2019.03.056). Excess renewable energy can be stored in the form of hydrogen energy and converted into electric energy through FC when needed.
‘This is mainly due to the high cost of FC components, including bipolar plates (BPs).’ It is also well-known that the cost of catalysts and membranes is also very high (10.1016/j.rser.2015.07.157). I suggest adding some comparisons between them to highlight the significance of this work.
Ad. 3. The proposed changes and missing references were incorporated into text.
- Page 2, ‘Furthermore, polymer-carbon composites exhibit much lower electrical and heat conductivity due to the content of the non-conductive polymer matrix...’ Why do not adopt conducting polymers, such as polyanilines? I consider it may make the polymer-carbon composites have similar, or at least not much lower conductivity compared to metallic ones.
Ad. 4. We agree that conductive polymers can serve as conductive binder but as we focused on commercial BPP no one material use it. We add comment to the text to mention potential of polyanailine as coating of metallic BPP.
- Page 2, ‘There is a threshold concentration, however, above which the low wettability of the particles leads to poor binding with the polymer matrix [16].’ Why does wettability have a relation with polymer-filler binding? For membranes, there are also inorganic filler-organic polymer hybrid ones, and extremely high loading of fillers also leads to the cracking of the composites (10.1016/j.electacta.2021.138133). This is mainly due to the phase segregation between the organic and filler phases that finally leads to cracking, but not related to wettability since both hydrophobic and hydrophilic fillers demonstrated the same trend.
Ad 5. We thank to the referee for this comment. We agree that there is partial similarity between heterogeneous membranes and composite BPP. In the case of BPP non-permeable properties are required thus strong interaction filler-binder will form less wettable material. We add the reference to the text.
- Page 10, ‘It is thus clear that in the cases described, the skin layer needs to be removed prior to the construction of a fuel cell stack.’ In terms of conductivity, it is true that it is better to remove the skin layer before use in a fuel cell stack. But there should be some reasons for using that skin layer, for example, to improve the chemical stability of the BPs by sacrificing a bit the conductivity. I consider there should be some comprehensive discussions about whether it is suitable to really remove the skin layer by considering multiple influencing factors.
Ad. 6. The skin layer is formed during extrusion or molding process due to the higher content of binder on BPP surface. Therefore it is not desired layer on surface but product of manufacturing. We did not observed any change of chemical resistance in the case of BPP without skin layer. But its negative impact on electric conductivity is evident.
- Page 10, ‘Polyethylene sulphide is used as a binder for the hardest material (Eisen-huth PPS) as well as for the material with one of low hardness (Shin Etsu BPB 350). The harder material contains 13 % of binder and is made by compression, while the softer material contains 15 % and is made by extrusion. From the information available, filler properties seem to be responsible for the resulting hardness of the material.’ I cannot fully agree with these descriptions. I consider that it should be the manufacturing process and binder composition together to determine the hardness.It is too absolute to say that only filler properties is responsible for the hardness, just because one binder leads to both the highest hardness and lowest hardness composites. It must be noted that the treatment/preparation process is also different.
Ad. 7. We thank to reviewer for this comment. We agree that manufacturing process has significant effect on fnal properties of BPP. We add this statement to the text.
- Page 13, ‘Figure 9: Hydrogen permeability values for BPs tested. DOE 2020 and 2025 target values [20,21] for flexural strength are shown as a green and a purple line, respectively’. Why does flexural strength appear in a figure related to hydrogen permeability?
Ad. 8. We thank to reviewer for identification of this mistake. The Figure description is corrected now.
Round 2
Reviewer 2 Report
The authors have addressed all my concerns.